# A Decision-Making Algorithm for Concrete-Filled Steel Tubular Arch Bridge Maintenance Based on Structural Health Monitoring

**DOI:** 10.3390/ma15196920

**Published:** 2022-10-06

**Authors:** Chengzhong Gui, Weiwei Lin, Zuwei Huang, Guangtao Xin, Jun Xiao, Liuxin Yang

**Affiliations:** 1School of Civil Engineering, Institute of Disaster Prevention, Sanhe 065201, China; 2Department of Civil Engineering, Aalto University, 02100 Espoo, Finland; 3Highway Monitoring & Response Center, Ministry of Transport of the P.R.C., Beijing 100005, China; 4China Highway Engineering Consulting Group Co., Ltd., Beijing 100089, China; 5CCCC Second Highway Engineering Co., Ltd., Xi’an 710199, China; 6Department of Civil and Environmental Engineering, Waseda University, Tokyo 1698555, Japan

**Keywords:** structural health monitoring, bridge maintenance decision-making, degenerated ultimate loading-capacity state, finite element analysis, performance evaluation, parameter regression, parameter prediction

## Abstract

This study focuses on establishing a novel heuristic algorithm for life-cycle performance evaluation. Special attention is given to decision-making algorithms for concrete-filled steel tubular (CFST) arch bridge maintenance. The main procedure is developed, including the ultimate loading-capacity modeling of CFST members, multi-parameter selection, ultimate thresholds presetting based on the finite element method, data processing, crucial parameters determination among sub-parameters, multi-parameter regression, ultimate state prediction, and system maintenance decision-making suggestions based on the multi-parameter performance evaluation. A degenerated ultimate loading-capacity model of CFST members is adopted in the finite element analysis and multi-parameter performance assessment. The multi-source heterogeneous data processing and temperature-effect elimination are performed for the data processing. The key sub-parameters were determined by the Principal Component Analysis method and the Entropy-weight method. The polynomial mathematical model is used in the multi-parameter regression, and the ±95% confidence bounds were verified. The system maintenance decision-making model combines the relative monitoring state, the relative ultimate state by the numerical analysis, and the relative residual life of degenerated members. The optimal system maintenance decision-making suggestions for the bridge maintenance system can be identified, including the most unfavorable maintenance time and parameter index. A case study on a CFST truss-arch bridge is conducted to the proposed algorithms. The obtained results demonstrated that the crack width deserves special attention in concrete bridge maintenance. Additionally, these technologies have enormous potential for the life-cycle performance assessment of the structural health monitoring system for existing concrete bridge structures.

## 1. Introduction

Bridge health monitoring technologies have been widely used in structural engineering, and three main approaches: model-based, signal-based, and hybrid methods [1], are the standard methods to analyze the time-series monitoring data. The purpose of structural health monitoring (SHM) is to diagnose the classification, location, and significance of structural conditions using mechanistic analysis, monitoring techniques, and data analysis [2] for bridge maintenance and management. Nevertheless, there is the fact that an urgent need should be paid attention to for the efficient inclusion of structural health monitoring (SHM) data in structural assessment and prediction models [3]. Especially when a bridge is aging and deteriorating, the condition assessment will increases the need for condition assessment [4]. With the diversification of monitoring methods and the improvement of sensor monitoring technology, the accuracy of monitoring structural response has increased, but the need to adapt to realistic structural assessment becomes increasingly complex. For this, the primary purpose of this paper is to propose a novel bridge maintenance decision-making algorithm, to facilitate the combination of structural health monitoring algorithms and bridge maintenance models.

The performance deterioration of structural members is a vital issue to be considered in a structural durability assessment. An example can illustrate the hazards due to the column strength deterioration; it may cause the shift to a weak story mechanism and weak story collapse under the excitation of multiple strong ground motions [5]. Four degenerated functions [6] reflect various deterioration mechanisms, covering a wide range of linear steel corrosion, quadratic sulfate attack, square-root diffusion-controlled degradation, and exponential degradation. The situation will be more complicated for the concrete-filled steel tubular (CFST) members due to combining two materials and geometries. This is because the experimental failure mode for the CFST members was local outward buckling of steel tube and concrete crushing near the corrosion [7].

For the assessment research of structural health monitoring, the SHM systems mainly focus on tracing the structural behavior and condition of the long-span bridges over their lifetime [8]. The bridge health monitoring system has involved research on numerous structural damage detection and assessment methods. Barbara Heitner [9] established a link between structural health monitoring data and the level of corrosion damage. Baghalian [10] put forward a novel nonlinear acoustic health monitoring (NAHM) approach for loose bolt detection and assessment, which proposes a low-cost alternative to the conventional SHM systems. Ha [11] developed an estimation method that uses the displacement assurance criterion (DAC) and displacement-based index (DBI) to detect structural damage, in which the DAC index can show the structural degradation’s occurrence, and the DBI can be as a suitable indicator for damage localization. Mohamed [12] and Oh [13] proposed the Bayesian learning method for structural damage assessment of the phase I IASC-ASCE benchmark problem. Mahato [14] proposed a combined recognition method of the signal processing model based on the wavelet-Hilbert transform. Andò [15] suggested a novel wavelet multi-resolution methodology approach to evaluate and validate alerts provided by an early warning system (EWS) for structural monitoring implemented through low-cost multi-sensor nodes.

The structural maintenance decision making is an optimization problem because it is also nonlinear and extensive in scale [16]. The technical parameter indexes (including the structural condition and the component condition), the average daily traffic (ADT) (including the truck traffic and non-truck traffic per year or hour), and the sufficiency rating (SR, the remaining serviceability of a bridge) are the three most essential parameters in a bridge maintenance system (BMS), developed by the Texas Department of Transportation (TxDOT) [17]. In addition, the maintenance decision-making models will consider the life-cycle cost/benefit analysis (CBA) [18] or cost-effective analysis (CEA) [19] to control the economic cost and to utilize the bridge performance to the greatest extent in a BMS.

The evaluation model in the bridge health monitoring system should be combined with bridge structure maintenance. The essential target is to realize the transition from traditional EM and PM to performance-based or condition-based maintenance of in-service bridges [8]. Regarding the optimal maintenance strategies based on monitoring information, Orcesi [3] put forward a model based on lifetime functions for predicting the evolution in time of the reliability of deteriorating bridges. Ni [8] developed the SHM&MMS system for formulation/updating of structural component degradation models based on continuously monitored data to predict the aging- and environment-induced degradation trend in structural components for maintenance management life-cycle service period.

For multi-parameter evaluation, the Multi-Criteria Decision Making (MCDM) was a powerful process for selecting the most sustainable solution to reach a consensus among economic, social, and environmental impacts [20]. Generally, the MCDM methods can be classified into the Multi-Attribute Decision Making (MADM) method and the Multi-Objective Decision Making (MODM) method. The Complex Proportional Assessment (COPRAS) method [21], and the Technique for Order of Preference by Similarity to Ideal Solution (TOPSIS) method [22] were the typical processes for the MADM process. The traditional MADM methods have limitations, and they are not sufficient to resolve real problems, such as the independence of the criteria, the linear aggregation, or the provision of the best alternative among themselves. The MODM methods can be considered as an effective process to find multiple trade-off solutions [20]. From the above, the comprehensive life-cycle performance assessment can be carried out for structural health monitoring with the above parameter indexes and different evaluation methods. However, challenges to implement data-driven bridge effective operation and maintenance decision-making are the difficulty of proving effective, such as lack of standard data needs, lack of data integration, and lack of standard procedures [23]. In addition, some of the problems existing in the current assessment methods include:(1)The current evaluation specifications of bridge health monitoring or maintenance management [17,24] tend to adopt the unified limit evaluation value of the mechanical performance (such as the strain and displacement) or technical condition, according to the authors’ previous investigations. However, the variation of mechanical performance at different structural members’ cross-sections is not the same in the limit loading-capacity state. This situation appears to be a typical problem in most such applications, so it is not reasonable to evaluate the structural performance by the method;(2)Existing studies of the data-driven prediction methods [25] have relied on substantial sample data and repeated iterative calculations. Researchers must adjust the proportion between the sample data, the total sample, and other calculation parameters to improve prediction accuracy. However, this suffers from limitations due to two problems. One is that it will take too much time to finish the calculation; another is that the results calculated by different prediction algorithms are inconsistent. In this situation, an approximate solution should be optimal for maintenance decision making to simplify the calculation.

This paper outlines an evaluation algorithm for bridge maintenance decision making based on structural health monitoring. Three main parts will be investigated. Section 3 will introduce the life-cycle performance decay model of the concrete-filled steel tubular members. Section 4 focuses on the multi-parameter performance state evaluation model of concrete bridges and proposes the optimal maintenance decision-making algorithm of concrete bridge systems decay model and life-cycle assessment model. Finally, a concrete-filled steel tubular truss-arch bridge case to validate the reasonability of the above algorithms is selected in Section 5 and Section 6.

## 2. Research Objectives and Framework

### 2.1. Research Objective

The present investigation aims to establish a novel life-cycle performance evaluation model for the concrete-filled steel tubular (CFST) arch bridges, which can evaluate the structural performance by analyzing the structural health monitoring data on the bridge maintenance decision making.

The bridge maintenance decision-making process can be treated as an optimization issue by defining an objective function relevant to the degenerated ultimate loading capacity state of CFST members, the measured structural multi-parameter performances based on the structural health monitoring, and the ultimate state by numerical analysis. The process generates the potential to use optimization algorithms and swarm intelligence methods for the life-cycle performance evaluation.

### 2.2. Framework of Algorithms

The concrete-filled steel tubular truss arch bridge and its members were studied as the research object. The process includes parameter selection, multi-source heterogeneous data preprocessing (including decomposition temperature effect), ultimate load-carrying state determination based on finite element numerical calculation and degeneration model, selection of critical sub-parameters, parameter fitting regression, updating, and prediction, life cycle assessment analysis, and comprehensive assessment of bridge structure system. Moreover, a life-cycle assessment method, combining bridge health monitoring and maintenance decision-making, is proposed. Some preliminary steps have been undertaken to understand the evaluation process better (see Figure 1). The following aspects were addressed in detail.

#### 2.2.1. A Life-Cycle Performance Decay Model of Structural Members Is Established

The CFST members are selected as the research object. A life-cycle decay model for concrete bridge members is established by equivalent material properties and geometries of CFST, considering the freeze-thaw and corrosion effects. The process provides a research basis for multi-parameter performance prediction, evaluation, and optimal maintenance decision-making of concrete bridges. The degenerated ultimate loading-capacity state of CFST bridge structures can be determined by the decay model of members and the finite element model-driven analysis.

#### 2.2.2. The Multi-Parameter Performance State Evaluation Model of Concrete Bridges Is Established

Based on the in-site monitoring data of the CFST truss arch bridge, four parameters are selected for the CFST arch bridge, including the strains of the arch ribs and the pier columns, the vertical displacements of arch ribs, and the crack width of the bridge deck beam. The linear interpolation method is used in the multi-source heterogeneous data process (MSHDP). After that, the tested data are preprocessed considering the temperature effect.

The dimensionless parameter index of state evaluation is defined as the parameter data divided by the ultimate threshold after the MSHDP and temperature effect elimination processing. Furthermore, the key sub-parameters can be selected by the Principal Component Analysis (PCA) method.

The polynomial curve (with 95% confidence bounds) is used to approximately describe the variation of the parameter index in parameter fitting regression, updating, and prediction.

Thus, the life-cycle assessment of a single parameter for the CFST truss arch bridge can be performed by parameter selection, degenerated ultimate threshold determination, data preprocessing, parameter regression, updating, and prediction.

#### 2.2.3. The Maintenance Decision-Making Algorithm of Concrete Bridge Systems Based on the Decay Model and Life-Cycle Assessment Model Is Partially Established

Based on the performance decay model of CFST truss-arch bridge members and the multi-parameter performance evaluation model of concrete bridges, the ultimate preset threshold, and the most unfavorable maintenance time and parameter indexes of the concrete bridge can be proposed.

## 3. Degenerated Ultimate Loading-Capacity State of CFST Members

### 3.1. Equivalent Material Properties and Geometries of CFST

#### 3.1.1. Equivalent Materials of CFST

For the confined CFST with the external diameter *D_s_* and thickness *t_s_* of the steel tube, the diameter of the core concrete can be calculated by the expression $D_{c}=D_{s}-2t_{s}$. Four stages [26] can be divided to determine the material model of the CFST in Figure 2. Two curves, including the ultimate compression strength of the unconfined concrete and confining concrete in CFST members, are used to determine the ultimate load-carrying capacities.

(1)For the OA stage

The ultimate compression strength of the confining concrete [27] can be determined as follows:(1)$$\sigma_{c\_OA}=0.4f_{c}^{\prime }$$
where $f_{c}^{\prime }$ represents the ultimate compression strength of the unconfined concrete.

(2)For the AB stage

The compressive strength of the unconfined concrete $\gamma_{fc}^{\prime }$ can be determined by the strength reduction factor $\gamma_{c}$ and the ultimate strength $f_{c}^{\prime }$, denoted as $\gamma_{fc}^{\prime }=\gamma_{c}f_{c}^{\prime }$, where $\gamma_{c}=1.85{\left(D_{s}-2t_{s}\right)}^{-0.135}$ [28], ranging from 0.85 to 1.0, considering the effects of the column size, the quality of concrete, and the loading rates on the concrete compressive strength [29].

The corresponding compressive strain of the unconfined concrete ($\varepsilon_{c}^{\prime }$) can be described as
(2)$$\varepsilon_{c}^{\prime }=\{\begin{array}{ll} 0.002 & \gamma_{fc}^{\prime }<28\\ 0.002+\left(\gamma_{fc}^{\prime }-28\right)/54000 & 28\leq \gamma_{fc}^{\prime }<82\\ 0.003 & 82\leq \gamma_{fc}^{\prime } \end{array}$$

Based on the calculation equations of the unconfined concrete, the compressive strength of the confined concrete ($f_{cc\_B}^{\prime }$) and the corresponding strain ($\varepsilon_{cc\_B}^{\prime }$) [29] can be expressed as
(3)$$f_{cc\_B}^{\prime }=\gamma_{c}f_{c}^{\prime }+k_{1}f_{rp}$$
(4)$$\varepsilon_{cc\_B}^{\prime }=\varepsilon_{c}^{\prime }\left(1+k_{2}f_{rp}/\left(\gamma_{c}f_{c}^{\prime }\right)\right)$$
where *k*_1_ = 4.1 and *k*_2_ = 5*k*_1_. The lateral confining pressure *f_rp_* [29] of the CFST can be determined by the nominal strength *f_y_* and the diameter thickness ratio *D_s_*/*t_s_*.
(5)$$f_{rp}=\{\begin{array}{cc} \left(0.02663-0.0002504D_{s}/t_{s}\right)f_{y} & 17\leq D_{s}/t_{s}<47\\ \left(0.01588-0.0000149D_{s}/t_{s}\right)f_{y} & 47\leq D_{s}/t_{s}<221 \end{array}$$

The AB stage of the constitutive model [30] is represented as
(6)$$\sigma_{c\_AB}=\frac{f_{cc\_B}^{\prime }\lambda (\varepsilon_{c}/\varepsilon_{cc}^{\prime })}{\lambda -1+{\left(\varepsilon_{c}/\varepsilon_{cc}^{\prime }\right)}^{\lambda }}$$
where the parameter $\lambda =\frac{E_{c}}{E_{c}-(f_{cc\_B}^{\prime }\;\;/\varepsilon_{cc\_B}^{\prime }\;\;)}$, *E_c_* stands for the tested Young’s modulus. When there are no experimental data on the concrete, *E_c_* can be determined by the estimation expression $3320\sqrt{\gamma_{c}f_{c}^{\prime }}+6900$ [31].

(3)For the BC stage:

Suppose the confining strain at Point C $\varepsilon_{e\_C}=10\varepsilon_{c}$, the degenerated BC stage of the constitutive model [30] is represented as
(7)$$\sigma_{c\_BC}=f_{cc\_B}^{\prime }+\frac{\varepsilon_{c}-\varepsilon_{cc\_B}}{\varepsilon_{e\_C}-\varepsilon_{cc\_B}}(f_{e\_C}-f_{cc\_B})\;\varepsilon_{cc\_B}<\varepsilon_{c}\leq \varepsilon_{e\_C}$$
where the confining strength at Point C $f_{e\_C}=\alpha_{c}f_{cc\_B}^{\prime }$, $\alpha_{c}=\{\begin{array}{cc} 0.8987-0.00122\gamma_{fc}^{\prime } & \gamma_{fc}^{\prime }<50\\ 0.774-0.0016\gamma_{fc}^{\prime } & \gamma_{fc}^{\prime }\geq 50 \end{array}$.

(4)For the CD stage:

Suppose the extreme strain at Point D $\varepsilon_{cu\_D}=30\varepsilon_{c}$, the extreme CD stage of the constitutive model [26] is represented as
(8)$$\sigma_{c\_CD}=f_{e\_C}^{\prime }+\frac{\varepsilon_{c}-\varepsilon_{e\_C}}{\varepsilon_{cu\_D}-\varepsilon_{e\_C}}(f_{e\_C}-f_{cc\_B}^{\prime })\;\varepsilon_{e\_C}<\varepsilon_{c}\leq \varepsilon_{cu\_D}$$
in which $\beta_{c}=\{\begin{array}{cc} \alpha_{c}-0.1 & \gamma_{fc}^{\prime }<50\\ 0.4 & \gamma_{fc}^{\prime }\geq 50 \end{array}$, $f_{cu\_D}=\beta_{c}f_{cc\_B}^{\prime }$.

#### 3.1.2. Equivalent Geometries of CFST

The equivalent compression and flexible stiffness parameters (*E_sc_A_sc_*, *E_sc_I_sc_*) of the CFST structure are determined by Equation (9) according to the Chinese specification for the design and construction of the concrete-filled steel tubular structures [32,33].
(9)$$\{\begin{array}{l} E_{sc}A_{sc}=E_{s}A_{s}+E_{c}A_{c}\\ E_{sc}I_{sc}=E_{s}I_{s}+E_{c}I_{c} \end{array}$$
where *E_s_*, *E_c_*, *A_s_*, *A_c_*, *I_s_*, and *I_c_* stand for the elastic modulus of steel tube and its core concrete, the sectional area, and the moment of inertia along its centroid axis, respectively.

### 3.2. Degenerated Ultimate Loading-Capacity State

The strain and displacement state generally affect the ultimate load-bearing capacities of the CFST members or bridge structures. When the strains of the structural members reach the material limit value or the load-displacement curve slope of the most unfavorable section approaches 0, the current state is considered the ultimate load-bearing state of the structure. In addition, the ultimate loading-capacity state should consider the time-variation deterioration of the structural performance. Four deterioration stages can be divided into the initial perfect stage, the propagation stage, the acceleration stage, and the deterioration stage. For the second stage, the initiation of the steel corrosion until cracking due to corrosion. For the third stage, the steel corrodes at a high rate as a result of cracking due to corrosion. Furthermore, for the ultimate stage, the load-bearing capacity is reduced considerably due to the increase in the corrosion amount.

The dimensionless degeneration parameter *g*(*N_c_*, *D_w_*, *f_c_*) of the CFST member under freeze–thaw cycles and corrosion, referenced from the work in [34] based on the specification Eurocode 4 [35], can be described as
(10)$$g(N_{c}(t),D_{w}(t),f_{c})=k_{sd,ft}k_{sr,co}$$
where $k_{sd,ft}$ stands for the degradation coefficient of the circular thin-walled CFST concrete strength *f_c_*, considering the influence of the number *N_c_*(*t*) of freeze–thaw cycles and concrete grades, proposed as $k_{sd,ft}=(1-0.0005N_{c}(t))(1+(f_{c}-30)/700)$. $k_{sr,co}$ stands for the strength degradation coefficient, considering the influence of corrosion rate *D_w_* after freeze–thaw, proposed as $k_{sr,co}=1-0.0015D_{w}(t)$.

The deterioration rate *D_w_*(*t*) can be defined as the weight loss of the steel tube (steel coupon) before and after the corrosion, respectively [34]. The average deterioration rate per year *ξ* is recommended in this study to simplify the weight loss calculation due to the actual steel corrosion, expressed as
(11)$$D_{w}(t)=W_{0}/W_{1}(t)-1=\xi \cdot t$$
where *W*_0_ and *W*_1_ are the weight of the steel tube (steel coupon) before and after the corrosion, respectively, the average degeneration rate *ξ* is set as 1 percent per year.

The bridge’s freeze–thaw number *N_c_*(*t*) relates to the freeze–thaw frequency per unit cycle *η* and the operation period *t*, which is expressed as
(12)$$N_{c}(t)=\eta \cdot t$$

Based on the statistics of the average high temperature and low temperature of the historical weather data of the in situ bridge condition, the freeze–thaw frequency of the bridge case can be determined by four times per year, *η* = 4. For the operation period, *t* = 9 years, a bridge member’s freeze–thaw number *N_c_*(*t* = 9) can be set as 36 times. Then, the relationship can be drawn as in Figure 3, consisting of the dimensionless deterioration factor, the operation period, and the strength of core concrete. When *f_c_* = 50.6 MPa, *g*(36, 9, 50.6) = 0.997.

The following expression can determine the deteriorated confined strength $f_{sc,g}(t)$ of the CFST member,
(13)$$f_{sc,g}(t)=f_{cc\_B}^{\prime }\cdot g(N_{c}(t),D_{w}(t),f_{c})$$

Based on the above strength, the axial ultimate loading capacity of the selected circular CFST members can be calculated by the expression $f_{sc,g}(t)\cdot \pi D_{sc}^{2}/4$, the recommended method by Gao [34], or the specification design method [35]. For the CFST arch bridge structure, the ultimate load-capacity state is complex to determine due to axial force, bending moment, torsion, and combination. Therefore, the finite element method (FEM) should be adopted.

### 3.3. Verification of Ultimate Loading-Capacity State

Twelve specimens were selected in the present study to compare with the previous experiment research [34] and verify the reasonable calculation of the ultimate loading-capacity state of the CFST member. The named specimens consisted of the concrete grade number, the freeze–thaw cycles per unit time, and corrosion rate. For example, S50-270-20-2 in Table 1 represents the second tested specimen filled with the C50 grade concrete under the environmental conditions: 270 freeze–thaw cycles and 20% corrosion rate.

From the literature [34], the tested CFST specimens were 270 mm long, and the external diameter-thickness ratio of the thin-walled steel tubes was 90/1.92. The Young’s modulus *E_s_* of the steel tubes were 2.01 × 10^5^ MPa, the strength ratio between the nominal yield and ultimate strength was *f_y_*/*f_u_* = 359/531, and the ultimate strain *ε_u_* of the steel tube was 0.3 *ε*. The nominal compressive strengths of the experimental specimens, in which steel tubes were filled with the C30, C40, and C50 grade concretes, were *f_c__*_30_ = 37.2 MPa, *f_c__*_40_ = 49.3 Mpa, and *f_c__*_50_ = 56.1 Mpa, respectively. Correspondingly, those Young’s modulus values were *E_c__*_30_ = 3.03 × 10^4^ MPa, *E_c__*_40_ = 3.26 × 10^4^ MPa, and *E_c__*_50_ = 3.37 × 10^4^ MPa, respectively.

By Equation (9), the equivalent solid diameter of the CFST cross-section is 0.1014 m. The equivalent Young’s modulus of the confined CFST specimens is 384,775 MPa, the sectional moment of inertia is 5.19 × 10^−6^ m^4^, and the equivalent mass density is 2947 kg/m^3^. From (1) ~ (8), the four-stage constitutive relation of three CFST materials, S30-0-0, S40-0-0, and S50-0-0, without the environmental effects of the freeze–thaw and corrosion, are depicted in Figure 4. A_1_~D_3_ represent the characteristic points of four-stage constitutive relation of three CFST materials, S30-0-0, S40-0-0, and S50-0-0.

The expressions $f_{sc,1}=(1.212+B\theta +C\theta^{2})f_{c}^{\prime }$, $\theta =A_{s}f_{y}/A_{c}f_{c}^{\prime }$, $B=0.176/f_{y}+0.974$, $C=-0.104f_{c}^{\prime }/14.4+0.031$ can calculate the ultimate compressive strength of the confined CFST specimens in the Chinese specification [33]. Thus, the equations can be used to compare with the ultimate strength $f_{cc\_B}^{\prime }$ of those members in Figure 4. The maximum difference Δ*f_sc_*_,1_ = (*f_sc_*_,1_ − *f’_cc_B_*)/*f_sc_*_,1_ between the two ultimate strengths is 7.899%, which indicates that they are similar to each other. However, the former specification does not consider freeze–thaw cycles and corrosion.

The expression $f_{sc,g}(t)\cdot \pi D_{sc}^{2}/4$ can calculate the axial ultimate loading capacity of the selected CFST members. The predicted value *N_up_* and the test value *N_ut_* [34] are compared with the present calculation *N_ul_* in Table 1 by calculating two differences Δ*N_up,ul_* = (*N_up_* − *N_ul_*)/*N_up_* Δ*N_ut,ul_* = (*N_ut_* − *N_ul_*)/*N_ut_*. The maximum difference between the first two specimens is 12.7%. Additionally, the maximum difference between the other specimens is 5.61%. Therefore, the equivalent modeling method in the present calculation is adequate to determine the ultimate loading-capacity state of the CFST members or bridge structures.

## 4. Multi-Parameter Performance Evaluation of CFST Bridges

The multi-parameter performance evaluation of the whole CFST bridge can be performed in four parts: parameter selection, ultimate thresholds presetting, data preprocessing, parameter regression, parameter prediction, and system maintenance decision-making processing.

### 4.1. Multi-Parameter Selection, Ultimate Thresholds Presetting, and Data Preprocessing

#### 4.1.1. Multi-Parameter Selection

There are multiple parameters to consider for the CFST arch bridges. From the perspective of the buckling analysis of arch structures, the arch ribs buckled in the first lateral mode, and the buckling modes of the arch ribs ranged from a single half-wave C-shape to a one-wave S-shape between the connections at the end [36]. Moreover, in the long term, the pin-ended and fixed CFST arch ribs may buckle in a symmetrical limit point instability mode or an antisymmetrical bifurcation mode [36]. The buckling modes vary obviously at the arch foot, 1/4 (3/4) of the arch rib’s span length, and the mid-span. Thus, the structural performance at the mid-span cross-section of arch ribs should be considered. The crack width of the concrete girder due to the combined external and initial loading action, which includes the bending stress, shear stress, or torsion stress, is generally divided into four intervals, 0.05 mm, 0.10 mm, 0.15 mm, and 0.2 mm [37], for building structures and bridge structures.

Herein, four parameter indexes are selected for the CFST arch bridge, including the strains of the arch ribs and the pier columns, the vertical displacements of arch ribs, and the crack width of the bridge deck beam, see Table 2. The time and the parameter index are mathematically noted as $t_{i,j}$ and $y_{i,j}(t)$, where the subscript *i* stands for the *i*-th time point i=1,2,…,n, *n* is the total time point, and the subscript *j* stands for the *j*-th parameter index, j=1,2,…,m, *m* is the total number of multi-parameter indexes.

#### 4.1.2. Ultimate Thresholds of Parameter Indexes

The ultimate parameter threshold should be determined to judge whether the structure is abnormal or the severity of the abnormality by the following methods:The historical data-driven method;The most unfavorable conditions by model-driven method;The standardized threshold.

Herein, the model-driven method is adopted to determine the ultimate threshold of structural parameters (the strains of arch ribs and pier columns, the displacements of arch ribs, and crack widths of deck beams). Furthermore, the finite element model (FEM) should be calculated beforehand. The specification [37] limits the maximum crack widths of deck beams.

The model-driven method can determine the maximum limit displacements and strains by calculating and extracting the structural limit state through numerical modeling. As engineers always install the sensors in a bridge health monitoring system during the structural service stage, the ultimate state of the bridge structure calculated by the FEM method cannot be directly used as the early warning threshold. Thus, the initial structural state before the sensor installation should be deducted from the ultimate state as the early warning threshold of the structural limit state. The traffic on the bridge deck is constantly interrupted during the construction for the sensors’ installation to ensure the reliability and validity of the data, so the initial structural state before the data collection or sensor installation can be considered as the initial structural state due to the structural self-weight
(14)$$T_{i,j}(t)=S_{Ultimate}(i,j)\cdot g(N_{c}(t),D_{w}(t),f_{c})-S_{G}(i,j)$$

In terms of the crack width of the bridge deck system, the crack width *w_c_* of the concrete girder due to the bending stress, shear stress, or torsion stress is generally divided into four intervals: 0.05 mm, 0.10 mm, 0.15 mm, and 0.2 mm [37] for the building structure or the bridge structure. Therefore, the ultimate crack width is set as 0.2 mm.

#### 4.1.3. Multi-Source Heterogeneous Data Processing

The Multi-Source Heterogeneous Data Analytic [38] processing methods include the linear interpolation approach, the spline interpolation approach, and the piecewise cubic Hermite interpolation polynomials (PCHIP) interpolation approach, as well as the Nearest-neighbor interpolation approach [39].

For the linear interpolation approach, the basic idea is to link two known points P_1_(*t_i_*_,*j*_, *y_i,j_*), P_2_(*t_i_*_,*j*+1_, *y_i,j_*_+1_) in a straight line at two adjacent times *t_i_*_,*j*_, *t_i_*_,*j*+1_ to be solved. During the adjacent time interval *t*_2_-*t*_1_, the corresponding dependent variable *y_i_* at any time *t_i_* can be determined as Equation (15) by the linear interpolation method.
(15)$$y_{i,j}(t_{k})=\frac{y_{i,j+1}-y_{i,j}}{t_{i,j+1}-t_{i,j}}(t_{k}-t_{i,j})+y_{i,j}$$
where *y_i,j_* can be considered the general definition of the response variables, including the strains, displacements, and crack widths.

#### 4.1.4. Data Preprocessing for Eliminating Temperature Effect

The temperature effect should be eliminated from the tested strains due to its significant influence on the variation of structural performance. The temperature effect considers the thermal diffusivity and thermal expansion. The basic principle of the former thermal diffusivity is to transfer the thermal energy from the higher temperature side to the lower temperature side of the structural cross-section, resulting in a thermal gradient that creates a gradient variation of strain. Furthermore, the later thermal expansion is related to the dilatation coefficient *α*, which can also be considered as the thermal gradient equals 1.0. Thus, the strain variation Δ*ε_T_*(*t*) because of temperature variation can be expressed as
(16)$$\Delta \varepsilon_{T}(t)=\alpha \cdot \Delta T(t)$$
where *α* is the dilatation coefficient [°C^−1^], and Δ*T* is the temperature variation [°C]. For concrete and steel, the dilatation coefficient *α* equals 10^−5^ °C^−1^.

Then, the tested strains $\varepsilon_{test}(t)$ will be modified as $\varepsilon_{test}^{\prime }(t)$ due to the temperature effects.
(17)$$\varepsilon_{test}^{\prime }(t)=\varepsilon_{test}(t)-\Delta \varepsilon_{T}(t)$$

#### 4.1.5. Determination of Key Sub-Parameters

Many sensors collect data in different locations for the structural health monitoring system, and it is not easy to comprehensively evaluate bridge structure health state. These are issues that popular methods can readily tackle. For efficient decision making in identifying the essential parameters and selecting a well-balanced solution, the Pareto optimal solutions [40] and the Principal Component Analysis (PCA) solutions [41] can be obtained. The PCA method is adopted herein to choose the key sub-parameters and the Entropy-weight (EW) method based on the correlation evaluation.

Definition of dimensionless parameter index

Before selecting the key sub-parameters, the parameter data $y_{i,j}(t)$ after the processing of the multi-source heterogeneous data preprocessing and the elimination of temperature effect is divided by the ultimate threshold $T_{i,j}(t)$ to obtain the dimensionless parameter index $R_{y_{i,j}}(t)$ of the state evaluation, expressed as Equation (18).
(18)$$R_{y_{i,j}}(t)=y_{i,j}(t)/T_{i,j}(t)$$

Process of PCA and EW method

The principal component analysis (PCA) and entropy-weight (EW) methods are the practical and widely used methods of selecting the most critical components, especially for multi-parameters with the same data attributes. The PCA is a dimensionality reduction method of variables by outputting the linear combination of the observed multi-parameters [41]. The fundamental problem of the PCA method’s application lies in the following: when any two-column parameters are the same, the PCA method will be invalid, so the correlation coefficient must be verified first, especially when there are few rows in the time direction. Herein, we use the correlation coefficients related to the original sub-parameters as the sub-parameter indexes during the principal component analysis. Our previous research outlined the analysis process [41].

The Entropy-weight (EW) method is a weight ratio evaluation method based on the information entropy theory, which can determine the decision-making or evaluation results by the seriously affected attribute with the highest diversity of attribute data [42].

Table 3 shows the PCA and EW method’s modified process based on the correlation coefficients of the dimensionless parameter index $R_{y_{i,j}}(t)$. The first two processing steps of the PCA method are the same as those of the EW method.

### 4.2. Multi-Parameter Regression, Updating, and Verification

#### 4.2.1. Parameter Regression Analysis

The polynomial mathematical model used in regression fitting can be expressed as
(19)$$y_{r_{i,j}}(t)=\sum_{i=k}^{N_{ierm}+1}A_{i,j}^{k}t_{i,j}^{N_{term}+1-k}$$
where the subscript *i*, *j*, and *k* represent the *i*-th parameter index, the *j*-th time point, and the *k*-th fitted coefficient, respectively. $t_{i,j}$, $y_{r_{i,j}}(t)$ stands for the time point and the fitting result and $A_{i,j}^{k}$ represents the *k*-th fitting term of the total *N_term_*. Herein, the quadratic polynomial model is adopted, then *N_term_* = 2.

#### 4.2.2. Goodness-of-Fit Verification

The R-square method is used to validate the difference between the target objects and the regression fitting results, which takes the coefficient of determination [45] as the verification index. The R-square value can be thought of as the ratio of the regression sum of squares and the total sum of squares, namely, the square of the correlation between the fitted datasets and the real-time datasets, expressed as,
(20)$$R_{square,i}=\sum_{i=1}^{n}{\left(y_{i,fit}-{\bar{y}}_{i}\right)}^{2}/\sum_{i=1}^{n}{\left(y_{i}-{\bar{y}}_{i}\right)}^{2}$$
where *y*_*i*,*fit*_, ${\bar{y}}_{i}$, *y_i_* represents the fitted model estimates, the averaged real-time measured data, and the real-time measured data of the *i*-th parameter index, respectively. The coefficient of determination value *R_square_* varies from 0 to 1, with a value closer to 1 indicating that the model accounts for a more significant proportion of variance.

### 4.3. System Maintenance Decision-Making Suggestions Based on Multi-Parameter Performance Evaluation

#### 4.3.1. Most Unfavorable Maintenance Time of Parameter Indexes

Set $y_{r_{i,j}}(t)$ = −1 and 1, by solving the polynomial fitted regression model, two-time points with real solutions under two extreme states −1 and 1 are obtained, and the maximum value of each parameter index’s time point is selected, see Equation (21).
(21)$$t_{\max[-1,1],ij}=[\max(\frac{-A_{2,ij}\pm \sqrt{A_{2,ij}^{2}-4A_{1,ij}(A_{3,ij}+1)}}{2A_{1,ij}}),\max(\frac{-A_{2,ij}\pm \sqrt{A_{2,ij}^{2}-4A_{1,ij}(A_{3,ij}-1)}}{2A_{1,ij}})]$$
where the subscript *i* and *j* represent the *i*-th parameter index and the *j*-th time point, respectively.

Thus, the maximum value is substituted by the polynomial regression fitting model to verify whether the calculated result is −1 or 1 (see Figure 5).

If so, the maximum prediction time interval *t*_*p*max,*i*_ of the *i*-th parameter index will be selected from the three-time points, including the solved time points *t*_max[−1],*i*_ and *t*_max[1],*i*_ corresponding to the fitted results −1 and 1, and the *i*-th time points beyond the historical data period (*t_i_*–*t_i_*_0_) from the calculated time point *t_i_* to the initial time point *t_i_*_0_, see Equation (22);


(22)
$$t_{p\max,i}=\max(t_{\max[-1],i}\;,t_{\max[1],i}\;,t_{i}-t_{i0})$$


If not, it is considered that the real fitted regression solution does not meet the prediction requirements.

Based on the above prediction, the global maintenance time *t*_maint,*i*_ can be determined by the sum of the maximum prediction time interval *t*_*p*max,*i*_ (unit: day or others) and its corresponding initial service year *t*_initial,*i*_ (unit: year) of the bridge members, expressed as Equation (23),
(23)$$t_{p\max,i}=\max(t_{\max[-1],i}\;,t_{\max[1],i}\;,t_{i}-t_{i0})$$

For the CFST bridge without replacement or with reconstruction, the initial service year *t*_initial,*i*_ of all the components can be considered the same; otherwise, the year *t*_initial,*i*_ should be modified by the member replacement.

#### 4.3.2. Most Unfavorable Parameter Index of the Bridge Maintenance System

The mathematical model proposed herein is based on the appropriate objective function form of the heuristic algorithm [46] to maximize the likelihood of the most unfavorable parameter index of the bridge maintenance system. The model considers the relationships:(1)The relative monitoring state variable between the monitoring state and the limit state by the numerical model;(2)The relative residual life between the predicted ultimate maintenance time and the design service life of bridge elements;(3)The relative residual performance of the degenerated structural components.

The objective function can be expressed as
(24)$$\alpha *=\underset{\alpha }{\arg\max}f_{obj1}(\alpha )=\underset{\alpha }{\arg\max}(\frac{1}{m_{i}}\sum_{j=1}^{m_{i}}R_{y_{i,j}}^{2}(t)+(1-\frac{t_{\mathrm{maint},i}}{t_{\mathrm{design},i}})(1-g(N_{c}(t_{\mathrm{maint},i}),D_{w}(t_{\mathrm{maint},i}),f_{c}))$$
where *t*_design,*i*_ stands for the design service life of the *i*-th component. The design service life *t*_design,b_ of the prestressed concrete girder in the bridge deck system is set as 70 years, that *t*_design,c_ of the pier column is 70 years [47], and that *t*_design,a_ of the arch truss is 100 years [48]. *t*_maint,*i*_ is the predicted replacement/maintenance time of the *i*-th component, *g*(*N_c_*(*t*_main,*i*_), *D_w_*(*t*_main,*i*_), *f_c_*) represents the performance decay coefficient of the component corresponding to the replacement/maintenance time *t*_maint,*i*_.

The symbol ‘arg max’ represents a mathematical expression for the selected scheme that takes the maximum value of *n* parameters corresponding to Equation (24). The first item of Equation (24) is used to evaluate the averaged relative monitoring state variable between the monitoring state and the limit state. Furthermore, the second item of Equation (24) aims to assess the relative residual service life and residual performance strength.

## 5. Case Study

### 5.1. Case Profile

#### 5.1.1. Description of Bridge Case and Sensor Layout

The selected bridge case QYRB [49] in China is a concrete-filled steel tubular truss arch bridge (see Figure 6). The construction of the bridge began in August 2006 and finished in September 2009. Figure 6 shows the total layout of the bridge case, which contains the main truss arch frame, the pier column structure, and the deck structure. Moreover, the bridge deck structure is a prestressed concrete hollow slab system above the truss arch frame.

The main truss arch frame named the transverse dumbbell CFST truss [50] consists of eight CFST arch ribs (A1) in the upstream and downstream, linked by the concrete-filled batten plates and K-shape supports (A4, A5) in the lateral direction. The steel tube (A3) is used as a web member between the truss arch frame’s upper and lower arch ribs. The clear span of each arch ribs *L_a_* = 180 m, their rises *H_a_* = 36 m, and the geometric shape along the span direction is a catenary with an equal cross-section. The total lateral width of the arch frame *W_a_* equals 8.7 m from the upstream to the downstream. The central vertical height from the top arch rib to the down arch rib *H_s_*= 2.8 m, and the central lateral distance (*W_s_*_,1_, and *W_s_*_,2_) from the upstream to the downstream is 1 m and 7 m, respectively, that is *W_s_*_,1_ = 1 m and *W_s_*_,2_ = 7 m.

For the pier column structure, the piers upon the prominent arch ribs from the side-span to mid-span are numbered as P1–P7 in Figure 6b. The reinforced concrete column P1 is set up above the arch abutment, with a rectangular section size 1.8 m × 1.5 m, with three equally distributed tie beams along the piers’ height direction. The external diameter-thickness ratio of the CFST pier column P2 and P3 are 800/12, and that of the CFST pier column P4–P7 are 600/10. The geometric cross-section of the truss arch frames and piers are listed in Table 4.

The whole width of the deck structure *W_d_* = 12 m. The 1st and 14th side span length *L_d_* of each prestressed concrete beam is 16 m, that is, *L_d_*_,1_ = *L_d_*_,14_ = 16 m, and its middle span length is 13 m, that is, *L_d_*_,2_ = *L_d_*_,3_= … = *L_d_*_,13_ = 13 m.

Two types of measuring sensors have been set up in the arch ribs and the pier columns, including the digital level instruments and strain gauges. The marked ‘PD-*i*’, ‘PS-*i*’, ‘AD-*i*’ and ‘AS-*i*’ in Figure 6a represent the *i*-th sensor location of the vertical measuring displacements and axial strains of the pier column and the arch, respectively. The monitoring period of the bridge case was from 30 May 2018 at 21:00:56 to 25 August 2018 at 07:17:23, which adds up to 86.428 days, and the initially referred temperature was 20.71 °C.

#### 5.1.2. Equivalent Geometries and Material Properties

Table 4 adopts two materials in the bridge case, including the Q345qc and C50. The A5 arch ribs and P1 piers are made of Q345qc and C50, respectively. In addition, the other arch ribs and piers are CFST members, constructed from both two materials. The Young’s modulus *E*, the mass density *ρ*, the Poisson’s ratio *μ_r_*, the yielding strength *f_y_* and the corresponding strain *ε_y_*, and the ultimate strength *f_cu_* and the corresponding strain *ε_cu_* of the two material properties are shown in Table 5. The thermal expansion coefficient is preset as *α_T_* = 10^−5^ /K for normal-weight concrete.

The CFST’s constitutive model consisted of the Q345qc steel tube and the C50 grade concrete, *f_y_* = 345 MPa, $f_{c}^{\prime }$ = 50.6 MPa, *E_c_* = 34,500 MPa. The equivalent geometries and stress–strain constitutive relationship of three CFSTs with the diameter–thickness ratio *D_s_*/*t_s_* = 600/12, 700/12, and 800/12, are depicted in Table 6.

### 5.2. Numerical Modeling and Verification

#### 5.2.1. Finite Element Modeling

The structural analysis of the case bridge was performed by the finite element analysis software DIANA 10.4 [39]. Two finite elements were set up to simulate the numerical modeling, including the Plate-bending elements and the Class-II Beam Element. The former element is adopted in the arch bottom and the batten plate between arch ribs. The latter analyzes the classical two-node straight beam elements in the linear static and nonlinear load-bearing state analysis. The Class-II Beam Element’s lateral displacement depends on the cubic Hermite shape function, which is similar to the Bernoulli beam theory, according to the DIANA 10.4 user manual.

Table 6 lists the stress–strain material properties of the bridge case.

#### 5.2.2. Verification

The vertical modes of the CFST bridge case were selected to compare the calculation results in the current study and the previous research. The natural frequency of the whole bridge corresponding to the first-order positive symmetric vertical bending mode is 1.171 Hz (see Figure 7), which is closely similar to 1.148463 Hz [52], therefore, the results by the equivalent modeling method agree well with that by the previous calculation method. The ultimate loading-capacity state can be determined based on the numerical model.

### 5.3. Analysis of Ultimate Load-Carrying State

The ultimate bearing capacity of the bridge is calculated by increasing the vertical uniform load of the bridge deck. With the increased bridge deck load, the load-displacement and load strain of the arch rib and its upper column show a nonlinear variation. When the material of the arch bridge reaches the ultimate strain or even the descending section, the displacement or strain of the structure increases significantly under the same load factor. When the load–displacement or load–strain curve tends to be horizontal, the bridge reaches the ultimate bearing capacity state.

From Figure 8, for the vertical displacement of arch ribs, the maximum average vertical displacement under the ultimate static state is 6.234 m at the middle section AD-8 of the arch rib, and the minimum vertical displacement is 1.184 m at the cross-section AD-4 along the arch ribs’ 1/4 (or 3/4) length. For the arch strains, the maximum strain of the main arch rib is 9.60 × 10^−3^ *ε* at the cross-section AS-8 of the arch rib’s end, and the minimum strain is 4.82 × 10^−3^ *ε* at the middle section AS-2. For the pier columns’ strain, the maximum strain of the upper column is 4.80 × 10^−2^ *ε* at the pier section PS-7, and the minimum strain is 5.19 × 10^−3^ *ε* at the pier section PS-4 along the 1/4 (or 3/4) length of the arch ribs.

The FEM analysis calculates two load-carrying states, including the self-weight state and the ultimate load-carrying state. The average displacements AD-2~AD-8 in the initial self-weight state are 0.0009 m, 0.0013 m, 0.0031 m, 0.0071 m, 0.0109 m, 0.0132 m, 0.0141 m, respectively. After subtracting the initial displacements, the ultimate displacement and strain thresholds of the CFST arch bridge model’s arch ribs and piers are listed in Table 7.

By Equation (25) [39], the equivalent Von Mises component strain *ε_eq_* in the initial self-weight state are determined by the axial stresses *ε_xx_*, *ε_yy_*, *ε_zz_* and shear strains *γ_xy_*, *γ_yz_*, *γ_xz_*.
(25)$$\varepsilon_{eq}=\frac{2}{3}\sqrt{\frac{3}{2}\left[{\left(\varepsilon_{xx}-\varepsilon_{yy}\right)}^{2}+{\left(\varepsilon_{yy}-\varepsilon_{zz}\right)}^{2}+{\left(\varepsilon_{xx}-\varepsilon_{zz}\right)}^{2}\right]+\frac{3}{4}\left(\gamma_{xy}^{2}+\gamma_{yz}^{2}+\gamma_{xz}^{2}\right)}$$

Similarly, the ultimate strain thresholds of arch ribs and piers can be obtained by subtracting the equivalent Von Mises strains in the initial state (see Table 8).

## 6. Results and Discussions

### 6.1. Monitoring Parameter Datasets

#### 6.1.1. Original Datasets of Measuring Points and Multi-Source Heterogeneous Data Analytic Processing

The original datasets of measuring points include one temperature point on Section 4# on the arch ribs, six strain points on the arch ribs, five strain points at the bottom of the pier column, four crack-width points at the bridge deck beams, and five displacement points on the arch ribs (see Table 9). The starting time is set from 30 May 2018 at 23:06:26, and the latest ending time of these measuring points is 25 August 2018 at 07:17:23, which lasts 86.428 days. The total number of time points is 66,560.

The multi-source heterogeneous data processing (MSHDP) is successfully performed based on the real-time interval of these selected 21 points by the linear interpolation method (see Table 9). There is no need to consider the coherence of the data in the above multi-source heterogeneous data processing process from Table 9. However, it is undeniable that there will be some adverse effects if data coherence is not considered. At the same time, the sensor data may have a significant mutation, and the impact will include at least two issues: Firstly, it affects the general variation trend of data. In turn, the accuracy of regression fitting and the accuracy of prediction need to be confirmed; secondly, it affects the rationality of the life-cycle performance evaluation of the structural system and then affects the scientific judgment of maintenance decision-making. One important future direction of the data processing is identifying and modifying the original data due to the sudden variation of time-series data collected by the sensor.

#### 6.1.2. Parameter Preprocessing by Eliminating Temperature Effect

The original parameter indexes, including the strains of arch ribs and pier columns, the crack widths of deck beam, and the displacements of arch ribs, were measured by the in-situ sensors installed in the arch bridge case, which contains the temperature effect. The time-series temperature effect can be transformed into the corresponding thermal strains by Equation (16). For the crack widths of the deck beam and the displacements of arch ribs, it is very complicated to determine the corresponding crack widths and displacements resulting from the temperature effect. Thus, the measured strains of the arch ribs and pier columns were separated from the temperature effects by Equation (17). Before eliminating the temperature effects, the median filtering process was performed using the current data preprocessing procedure, which aims to eliminate the local peak (see Figure 9). By subtracting the above thermal strains, we can draw the calculation results of the original parameter indexes in Figure 10. Comparing Figure 9 with Figure 10, the results obtained here may have implications for understanding that the temperature effect should not be ignored. Further improvements are expected to result in an improved understanding of eliminating the temperature effect from the crack width and vertical displacements of the original parameter indexes.

#### 6.1.3. Selection of Key Sub-Parameter Based on Correlation Coefficients

The Principal Component Analysis evaluates the essential sub-parameters for multivariable parameters (PCA) and the Entropy-weight (EW) method. Both methods are based on the correlation coefficients between the original sub-parameter datasets (see Figure 11). Most of the correlation coefficients differ from each sub-parameter, demonstrating that the key sub-parameters for multivariable parameters can be determined not by the original datasets, but by their correlation coefficients.

For the CFST arch bridge case, the key sub-parameters for the arch strains, pier strains, crack widths of deck beams, and arch displacements are ‘AS-4’, ‘PS-6’, ‘1#’, and ‘AD-3’, respectively, by adopting the PCA and EW method, see Table 10. A satisfactory agreement is found when comparing results by two methods.

### 6.2. Results of Parameter Regression and Prediction

The polynomial regression fitting was performed for all the most important sub-parameters of the measuring points, and the R-Squared error (RSE) was calculated. The RSE value reflects that the model explains approximately 75% of the variability in the response variable.

Before the parameter regression, the relative values of all the key sub-parameters can be calculated by Equation (18) and used to fit the regression coefficients of parameter indexes. Table 11 and Figure 12 show the fitted regression coefficients and their coefficients with ±95% confidence bounds by Equation (19). In Table 11, FC, FC_−95%, and FC_95% represent the fitted coefficients, the corresponding coefficients with the −95% and 95% confidence bound, respectively, and the RSE values are provided. Based on the parameter regression model, the parameter predictions of four key sub-parameters have been performed to determine the ultimate state, see Figure 12. The ultimate service period *t*_max[−1,1]_ in Table 11 represents the ultimate prediction time corresponding to the relative object ultimate state −1 and 1. From Figure 12, most measured datasets are within the ±95% confidence bound.

Although the regression coefficients and their coefficients with 95% confidence bounds have been successfully fitted, it is undeniable that it should improve the fitted accuracies before conducting further extensive studies on the regression coefficients by trying different regression fitting models.

### 6.3. Results of the Life-Cycle Assessment Analysis

By substituting each value of the *t*_max[−1,1]_ into the polynomial fitted regression model, the ultimate service time *t*_max_ and the corresponding relative ultimate state −1 or 1 can be determined. The ultimate state can be considered Level 5 in the Chinese highway bridge maintenance specification [24]. It demonstrates that the prominent members have severe defects that endanger bridge safety and even reach the ultimate load-carrying state. The ultimate time of the total bridge system can be initially determined by the minimum ultimate time of the components. That is to say, the crack-width of the bridge deck beams should be considered first, and the maintenance measures should be taken for the bridge deck system in advance. We note, however, that the predicted maintenance priority of the other three-parameter indexes is the pier strain, arch displacement, and arch strain by the ultimate service time. Therefore, the most unfavorable maintenance time of the structural system should be 54 days from the initial monitoring time. The most unfavorable parameter index of the bridge maintenance system from the maximum to the minimum can be listed as the crack width of the bridge deck system, arch strain, arch displacement, and the pier column strain.

Considering Table 12, this study is the first step toward a more profound understanding of the heuristic algorithm for structural system maintenance decision making. It differs from the majority of other studies. However, some study limitations should be acknowledged that the optimization results may differ when adopting other regression fitting models and appropriate objective function of optimization criteria. The effectiveness of these optimization methods should be substantiated.

## 7. Conclusions and Further Study Work

This paper proposed a decision-making algorithm for concrete-filled steel tubular (CFST) arch bridge maintenance using model-driven and data-driven methods. The most relevant conclusions drawn are as follows:(1)A new multi-parameter performance evaluation approach for CFST bridge maintenance decision-making identification is proposed based on the heuristic algorithm, including the processes of the multi-parameter selection, ultimate thresholds presetting, data processing, multi-parameter regression, parameter prediction, and system maintenance decision-making suggestions;(2)A life-cycle performance decay model of CFST bridge components have been considered to determine the degenerated ultimate loading-capacity thresholds with the environmental effects of the freeze–thaw and corrosion;(3)The multi-source heterogeneous data processing and eliminating temperature effect are performed. The key sub-parameters can be determined by the Principal Component Analysis method and the Entropy-weight method. Specifically, the temperature effect should not be ignored to modify the original parameter indexes. Furthermore, the critical sub-parameters for multivariable parameters can be determined by the correlation coefficients of the original datasets;(4)The polynomial mathematical model is approximately used in the multi-parameter regression fitting with ±95% confidence bounds and verified by the goodness-of-fit of the R-square errors, which aims to develop a framework to investigate the future variation of the time-serial multi-parameter indexes;(5)The optimal system maintenance decision-making suggestions can be identified based on the multi-parameter performance evaluation, including the most unfavorable maintenance time and the parameter index for the bridge maintenance system. The algorithm relies on the regression fitting models and appropriate objective functions of optimization criteria;(6)A CFST truss-arch bridge case and measuring sensor layout are introduced for illustrative purposes, and the current numerical modeling determines the ultimate loading-capacity state. The crack width of the bridge deck system is the main problem for concrete bridges. Moreover, the results demonstrate that the multi-parameter performance evaluation confirms the maintenance decision-making process based on the ultimate numerical state and the enormous amount of measured data of multi-parameter indexes;(7)These technologies have enormous potential as the appropriate evaluation system can provide a perspective of life-cycle performance assessment of the structural health monitoring system for existing concrete bridge structures.

Potential weaknesses are loss of practical validation and difficulty in the effective pre-maintenance decision making, which is a potential area for further research, at least including the following aspects:(1)The main focus of our model will be on the relative parameter response interval of structural members according to regular and preventive maintenance;(2)Logistic regression analyses will assess the association between parameter indexes, providing more closely fitted results. Moreover, the appropriate objective functions of optimization criteria will be further validated;(3)Reliability-based safe factor will be defined as a parameter index during the multi-parameter performance evaluation, relevant to the life-cycle cost analysis.

## Figures and Tables

**Figure 1 materials-15-06920-f001:**
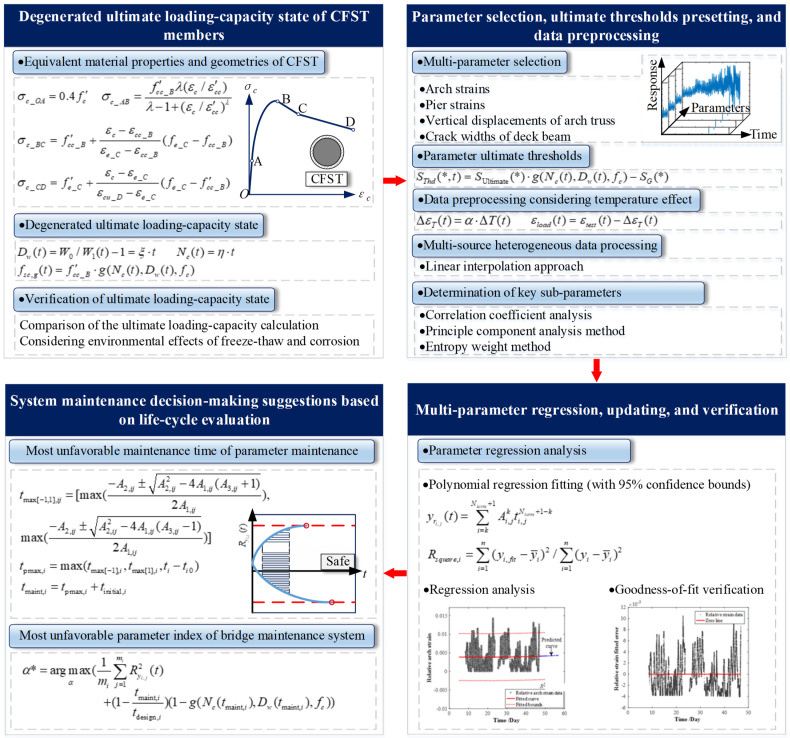
Procedure of life-cycle performance evaluation for CFST bridges.

**Figure 2 materials-15-06920-f002:**
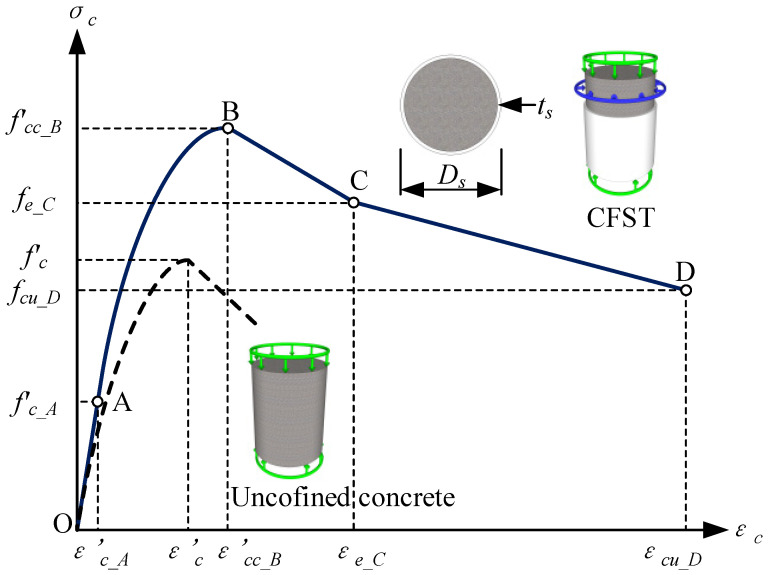
Stress–strain model for the CFST.

**Figure 3 materials-15-06920-f003:**
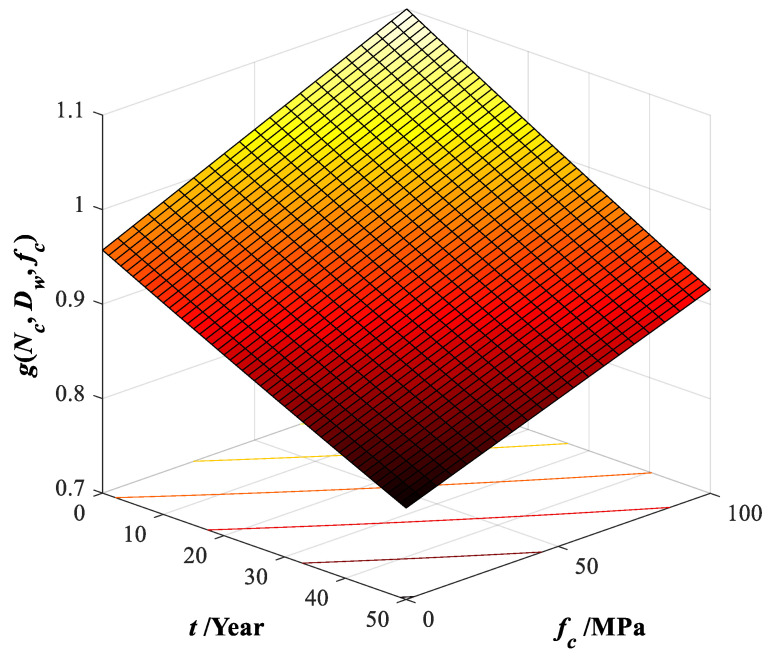
Dimensionless deterioration factor for the confined CFST.

**Figure 4 materials-15-06920-f004:**
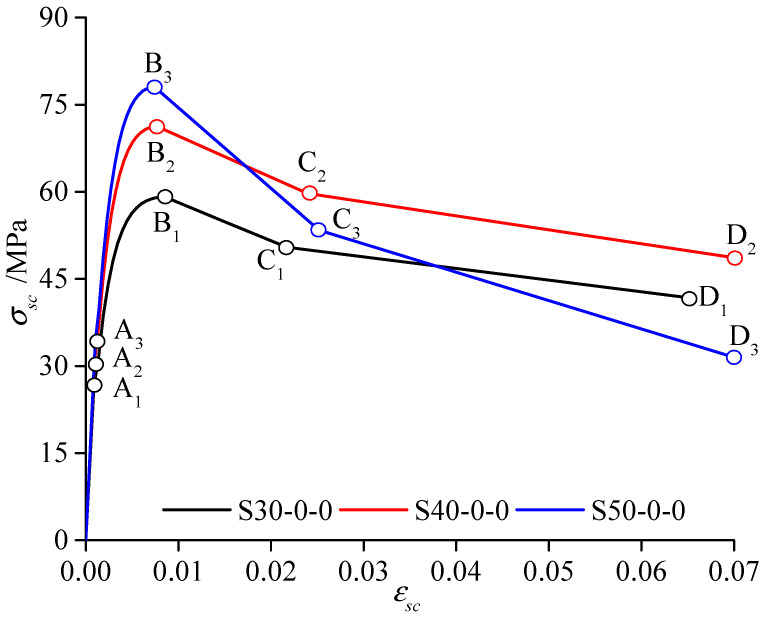
Stress–strain model for confined CFST.

**Figure 5 materials-15-06920-f005:**
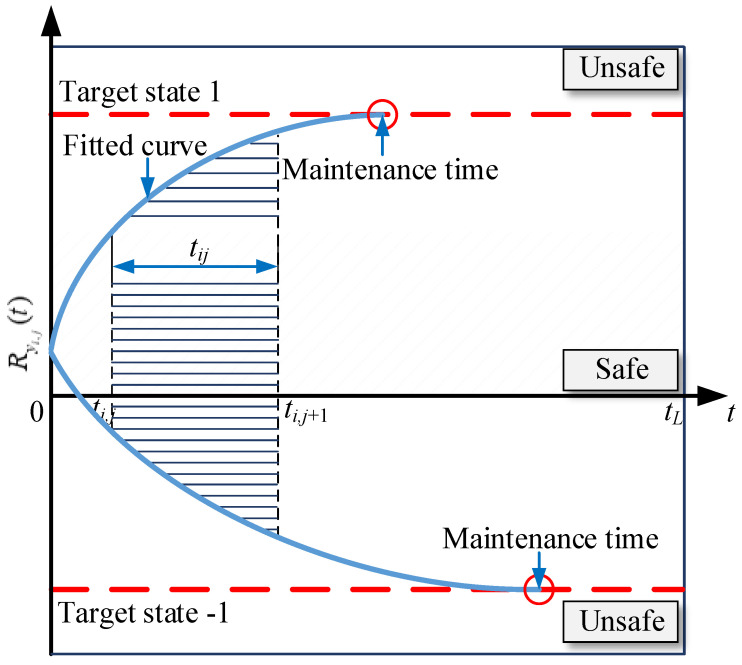
Determination diagram of the unfavorable maintenance time.

**Figure 6 materials-15-06920-f006:**
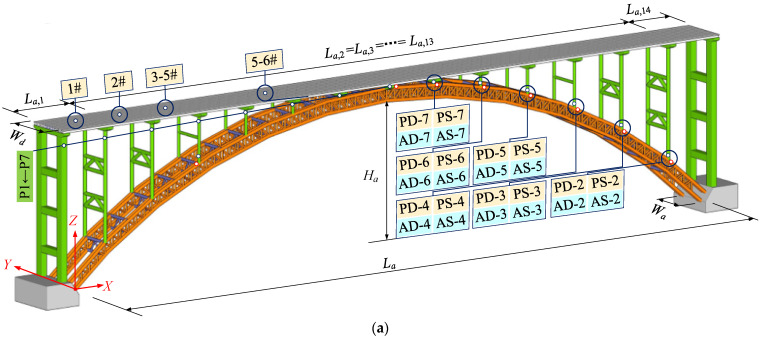
Layout diagram and sensor location of the bridge case. (**a**) Bridge layout. (**b**) Truss arch frame.

**Figure 7 materials-15-06920-f007:**
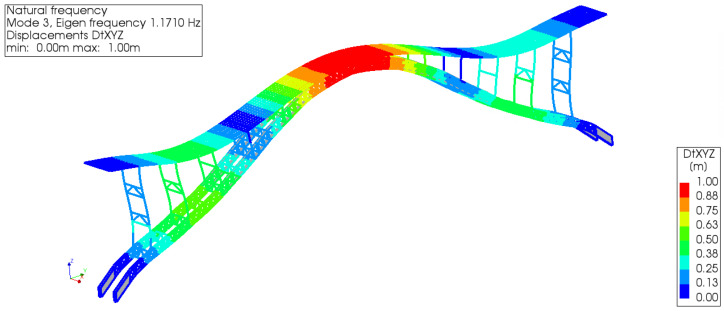
The first-order positive symmetric vertical bending mode of the bridge case by DIANA.

**Figure 8 materials-15-06920-f008:**
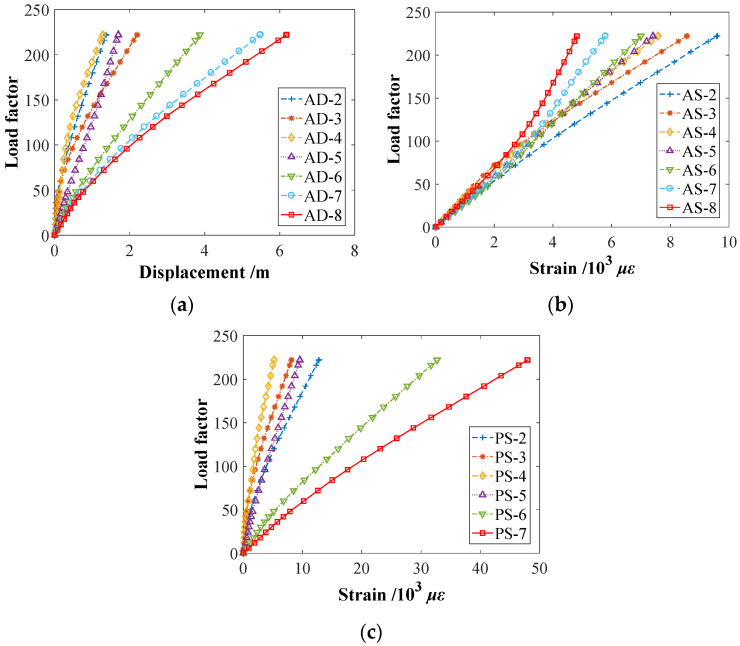
Numerical analysis of the load-carrying capacity. (**a**) Displacements of the upper arch ribs; (**b**) Strains of arch ribs. (**c**) Strains of pier columns.

**Figure 9 materials-15-06920-f009:**
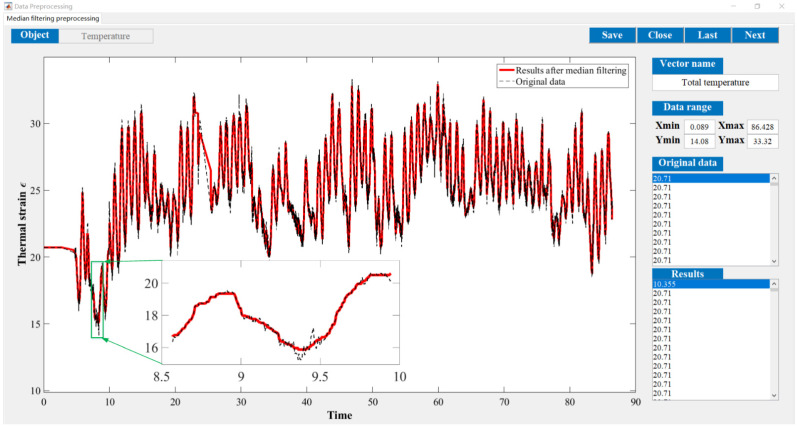
Thermal strains after the median filtering process.

**Figure 10 materials-15-06920-f010:**
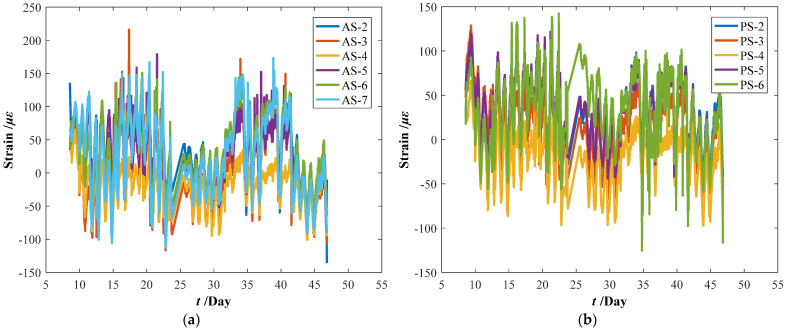
Measuring datasets of the bridge case considering eliminating temperature effect. (**a**) Strains of arch ribs. (**b**) Strains of pier column ends. (**c**) Crack widths of deck beams. (**d**) Displacements of arch ribs.

**Figure 11 materials-15-06920-f011:**
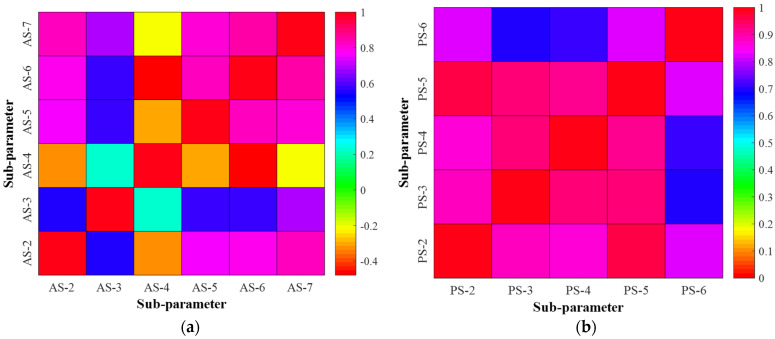
Correlation coefficient between monitoring sub-parameters. (**a**) Arch strain. (**b**) Pier strain. (**c**) Crack width of deck beam. (**d**) Arch displacement.

**Figure 12 materials-15-06920-f012:**
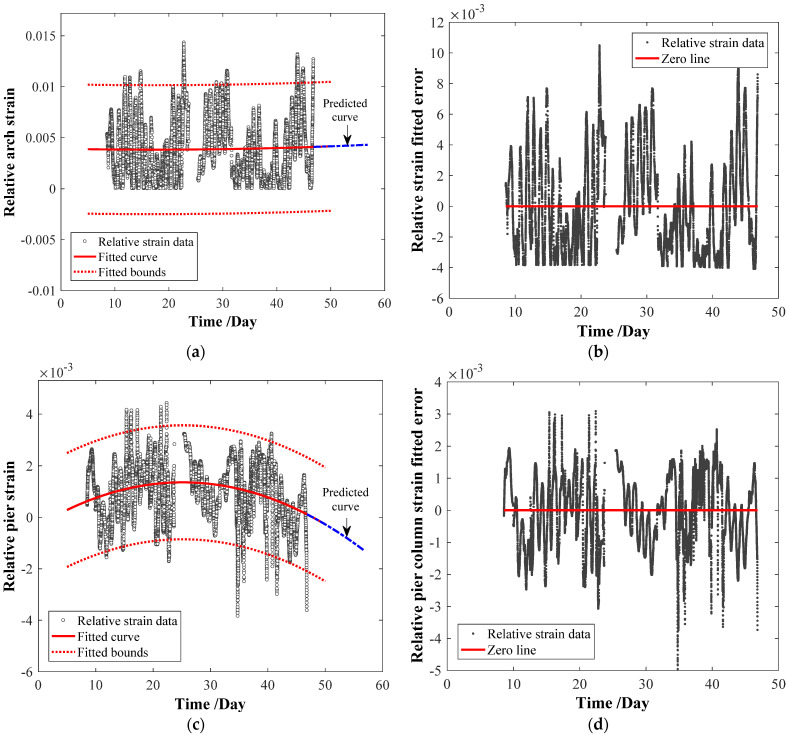
Parameter regression and prediction. (**a**) Relative arch strain. (**b**) Fitted error of arch strain. (**c**) Relative pier column strain. (**d**) Fitted error of pier column strain. (**e**) Relative crack width of deck beam. (**f**) Fitted error of crack width of deck beam. (**g**) Relative arch displacement. (**h**) Fitting error of arch displacement.

**Table 1 materials-15-06920-t001:** Comparison of the ultimate loading-capacity calculation.

Specimen	Previous Experiment Study			Present Study		Difference		
	*f_sc_*_,1_ [33] [MPa]	*N_up_* [34] [kN]	*N_ut_* [34] [kN]	*f’_cc_B_* —— [MPa]	*N_ul_* —— [kN]	Δ*f_sc_*_,1_ — [%]	Δ*N_up,ul_* —— [%]	Δ*N_ut,ul_* —— [%]
S30-0-0-1	60.88	543.60	538.80	59.12	482.33	2.89	12.70	11.71
S30-0-0-2	60.88	543.60	522.20	59.12	482.33	2.89	12.70	8.27
S40-0-0-1	72.74	603.60	614.10	71.22	590.99	2.10	2.13	3.91
S40-0-0-2	72.74	603.60	617.40	71.22	590.99	2.10	2.13	4.47
S50-0-0-1	84.71	637.30	659.10	78.02	653.54	7.90	−2.48	0.85
S50-0-0-2	84.71	637.30	653.10	78.02	653.54	7.90	−2.48	−0.07
S50-90-5-1	-	611.60	618.10	78.02	619.45	-	−1.27	−0.22
S50-90-5-2	-	611.60	619.50	78.02	619.45	-	−1.27	0.01
S50-180-10-1	-	586.60	599.80	78.02	585.8	-	0.14	2.39
S50-180-10-2	-	586.60	602.50	78.02	585.8	-	0.14	2.85
S50-270-20-1	-	551.10	579.10	78.02	548.35	-	0.50	5.61
S50-270-20-2	-	551.10	573.20	78.02	548.35	-	0.50	4.53

**Table 2 materials-15-06920-t002:** Multi-parameter selection.

No.	Parameters	Where to be Focused
1	The strain of the arch ribs	Arch foot, 1/4 (3/4) of arch rib’s span length, mid-span, and arch rib’s cross-section nearby the junction of an arch rib and column pier
2	The strain of the pier columns	Pier bottom nearby the junction of an arch rib and column pier
3	Vertical displacement of arch ribs	1/4 (3/4) of arch rib’s span length, mid-span, arch rib’s cross-section nearby the junction of an arch rib and column pier
4	Crack width of the bridge deck beam	Bridge deck beam cross-section

**Table 3 materials-15-06920-t003:** Steps of Principal Component analysis and Entropy-weight method.

No.	PCA Method [43]	EW Method [44]
1	Selecting the sub-parameter indexes $R_{y_{i,j}}(t)$	Selecting the sub-parameter indexes $R_{y_{i,j}}(t)$
2	Calculating the correlation coefficients of the sub-parameter indexes $C_{y_{i,j}}(t)$	Calculating the correlation coefficients of the sub-parameter indexes $C_{y_{i,j}}(t)$
3	Calculating the mean ${\bar{c}}_{i}=\sum_{j=1}^{N}c_{ji}$, and standard deviation $s_{i}=\sqrt{\frac{1}{N-1}\sum_{j=1}^{N}{\left(c_{ji}-{\bar{c}}_{i}\right)}^{2}}$	Calculating the proportion of the *i*-th time point in the *j*-th parameter index $p_{ij}=C_{y_{i,j}}/\sum_{i=1}^{n}C_{y_{i,j}}$
4	Calculating the covariance matrix $R=\left[r_{ij}\right]$, $r_{ij}=\frac{1}{N-1}\sum_{k=1}^{N}Y_{ki}Y_{kj}$, $Y_{ki}=(c_{ij}-{\bar{c}}_{j})/s_{j}$	Calculating the entropy of the *j*-th parameter index $e_{j}=-k\sum_{i=1}^{n}p_{ij}\ln(p_{ij})$, where k=1/ln(n)>0, which satisfies to $e_{j}\geq 0$
5	Calculating the eigenvalues $\lambda_{1},\lambda_{2},\;\ldots ,\lambda_{3}$ and eigenvector $l_{i}={\left[\begin{array}{cccc} l_{1i} & l_{2i} & \cdots & l_{pi} \end{array}\right]}^{T}$ of the sub-parameters	Calculating the information entropy redundancy $d_{j}=1-e_{j}$
6	Sorting the eigenvalues from maximum to minimum, checking the number *m* to satisfy $\sum_{j=1}^{m}\lambda_{j}/\sum_{j=1}^{p}r_{jj}>0.85$	Calculating the weight ratio of each parameter index, and checking the sample number *k* to satisfy $w_{k}=\max(d_{j}/\sum_{j=1}^{m}d_{j})$ related to the maximum weight

**Table 4 materials-15-06920-t004:** Geometries and material composition of the cross-sections.

Member	P1	P2~P3	P4~P7	A1	A2	A3	A4	A5
*D_s_* × *t_s_*	1.8 × 1.5	D800 × 12	D600 × 10	D700 × 12	D600 × 10	D325 × 8	D600 × 10	D299 × 8
Materials	C50	C50/Q345qc	C50/Q345qc	C50/Q345qc	C50/Q345qc	Q345qc	C50/Q345qc	Q345qc

**Table 5 materials-15-06920-t005:** Materials properties [51].

Material	*E* [10^4^ MPa]	*ρ* [kN/m^3^]	*μ_r_*	*f_y_* [MPa]	*ε_y_* [10^−3^]	*f_cu_* [MPa]	*ε_cu_* [10^−3^]
Q345qc	20.6	2500	0.3	405	1.966	540	200
C50	3.45	7850	0.2	-	-	49.75	2.2

**Table 6 materials-15-06920-t006:** Equivalent geometries and constitutive model for the confined CFST.

*D_s_* × *t_s_* [mm × mm]	*D_cs_* [m]	*E_cs_* [MPa]	*I_CS_* [10^−3^m^4^]	*ρ_CS_* [kg/m^3^]	*f’_c_A_* [MPa]	*ε’_c_A_* [10^−3^*ε*]	*f’_cc_B_* [MPa]	*ε’_cc_B_* [10^−3^*ε*]	*f_e_C_* [MPa]	*ε_e_C_* [10^−3^*ε*]	*f_cu_D_* [MPa]	*ε_eu_D_* [10^−3^*ε*]
600 × 10	0.665	37,199	9.619	2851	31.436	0.911	64.208	7.89	54.334	22.78	44.461	68.34
700 × 12	0.778	37,323	17.948	2861	31.436	0.911	64.243	7.9	54.364	22.78	44.485	68.34
800 × 12	0.881	36,778	29.616	2816	31.436	0.911	64.067	7.85	54.215	22.78	44.364	68.34

**Table 7 materials-15-06920-t007:** Ultimate displacement thresholds of arch ribs.

Location [m]	Vertical Displacement of Arch Ribs			
	Code	Ultimate State [m]	Initial State [m]	Modified Ultimate Threshold [m]
12	AD-2	−1.37	−0.001	−1.369
25	AD-3	−2.095	−0.001	−2.094
38	AD-4	−1.184	−0.003	−1.181
51	AD-5	−1.879	−0.007	−1.872
64	AD-6	−4.084	−0.011	−4.073
77	AD-7	−5.637	−0.013	−5.624
90	AD-8	−6.234	−0.014	−6.220

**Table 8 materials-15-06920-t008:** Ultimate strain thresholds of arch ribs and pier columns.

Location [m]	Strains of Arch Ribs				Strains of Pier Ends			
	Mark	Ultimate State [10^−3^ ε]	Initial State [10^−5^ ε]	Modified Ultimate Threshold [10^−3^ ε]	Mark	Ultimate State [10^−2^ ε]	Initial State [10^−5^ ε]	Modified Ultimate Threshold [10^−2^ ε]
12	AS-2	9.60	3.537	9.56	PS-2	1.28	1.34	1.28
25	AS-3	8.57	3.542	8.53	PS-3	0.814	1.48	0.813
38	AS-4	7.59	3.638	7.55	PS-4	5.19	2.30	0.517
51	AS-5	7.41	3.756	7.37	PS-5	0.955	5.88	0.949
64	AS-6	7.00	3.736	6.96	PS-6	3.27	12.8	3.26
77	AS-7	5.79	3.225	5.76	PS-7	4.80	19.3	4.78
90	AS-8	4.82	3.332	4.79	-	-	-	-

**Table 9 materials-15-06920-t009:** Original datasets before and after MSHDP.

Parameter	Datasets before and after MSHDP
Temperature	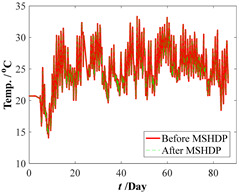
Arch strain	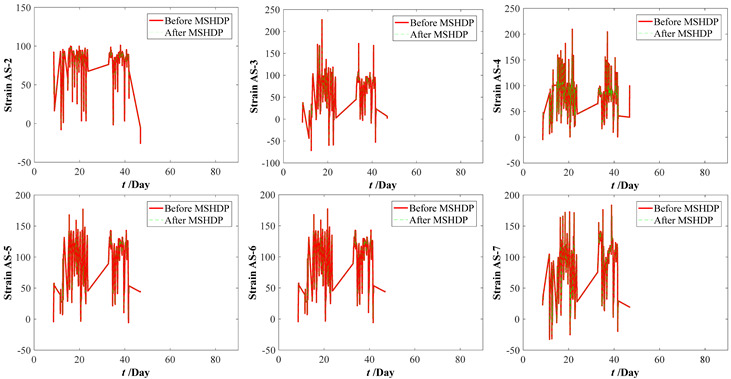
Pier strain	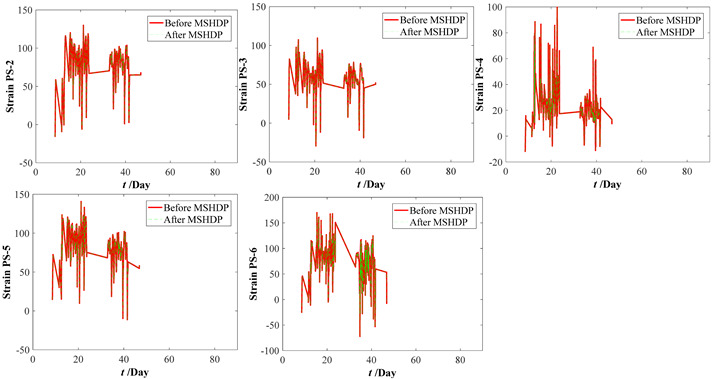
Crack width	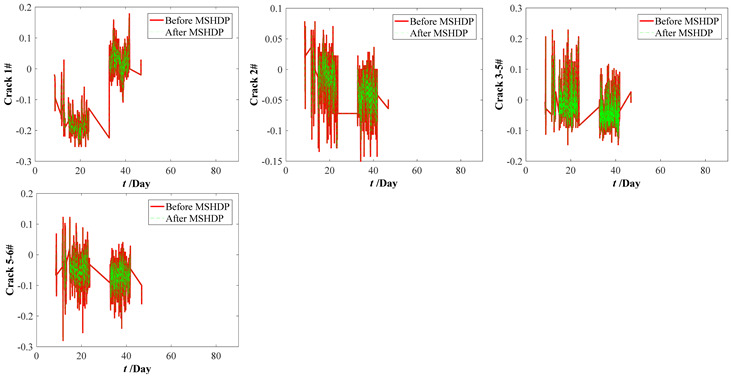
Arch displacement	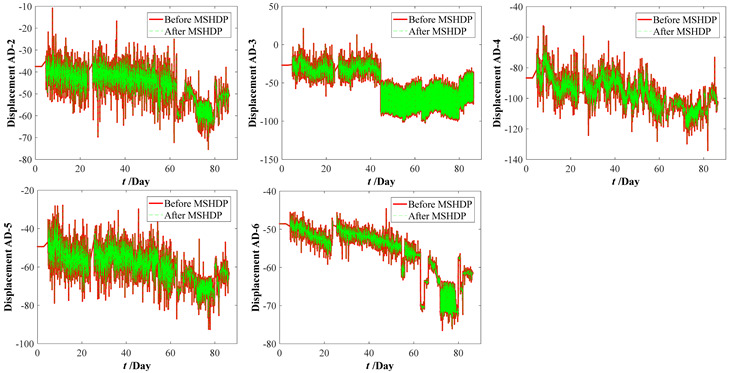

**Table 10 materials-15-06920-t010:** Key measuring point selection for multivariable parameters.

Method	Arch Strain	Pier Strain	Crack Width of the Deck Beam	Arch Displacement
PCA	AS-4	PS-6	1#	AD-3
EW	AS-4	PS-6	1#	AD-3

**Table 11 materials-15-06920-t011:** Fitted regression coefficients and their coefficients with ±95% confidence bounds.

Parameter		*A* _1_	*A* _2_	*A* _3_	RSE
Arch strain	FC	3.04 × 10^−7^	−1.04 × 10^−5^	3.921 × 10^−3^	6.466 × 10^−4^
	FC_−95%	−5.09 × 10^−8^	−3.06 × 10^−5^	3.662 × 10^−3^	
	FC_95%	6.58 × 10^−7^	9.86 × 10^−6^	4.180 × 10^−3^	
Pier strain	FC	−2.64 × 10^−6^	1.330 × 10^−4^	−3.148 × 10^−4^	0.0817
	FC_−95%	−2.77 × 10^−6^	1.259 × 10^−4^	−4.054 × 10^−4^	
	FC_95%	−2.52 × 10^−6^	1.400 × 10^−4^	−2.243 × 10^−4^	
Crack width	FC	1.314 × 10^−3^	−4.391 × 10^−2^	−4.612 × 10^−1^	0.5437
	FC_−95%	1.277 × 10^−3^	−4.600 × 10^−2^	−4.880 × 10^−1^	
	FC_95%	1.350 × 10^−3^	−4.181 × 10^−2^	−4.344 × 10^−1^	
Arch displacement	FC	−2.76 × 10^−6^	5.926 × 10^−4^	5.791 × 10^−3^	0.6398
	FC_−95%	−2.84 × 10^−6^	5.844 × 10^−4^	5.624 × 10^−3^	
	FC_95%	−2.67 × 10^−6^	6.008 × 10^−4^	5.958 × 10^−3^	

**Table 12 materials-15-06920-t012:** Results of system maintenance decision making.

Parameter		Predicted Ultimate Service Time			*t*_maint_ [Year]	*t*_design_ [Year]	*g*(*N_c_*(*t*), *D_w_*(*t*), *f_c_*)	Equation (24)
		*t*_max[−1,1]_ [Day]	*t*_max_ [Day]	State				
Arch strain	FC	[17.1, 1828.2]	1828.2	1	13.76	100	0.98	1.69 × 10^−2^
	FC_−95%	[4.1486, -]	-	1	-	100	-	-
	FC_95%	[-, 1222.5]	1222.5	1	12.1	100	0.986	1.21 × 10^−2^
Pier strain	FC	[640.69, 25.16]	640.69	−1	10.5	70	0.992	6.86 × 10^−3^
	FC_−95%	[624.23, 22.75]	624.23	−1	10.46	70	0.992	6.73 × 10^−3^
	FC_95%	[658.40, 27.79]	658.4	−1	10.55	70	0.992	7.01 × 10^−3^
Crack width	FC	[16.71, 54.01]	54.01	1	8.9	70	0.998	4.62 × 10^−1^
	FC_−95%	[18.01, 56.61]	56.61	1	8.9	70	0.998	4.62 × 10^−1^
	FC_95%	[15.49, 51.57]	51.57	1	8.89	70	0.998	4.62 × 10^−1^
Arch displacement	FC	[720.50, 107.36]	720.5	−1	10.72	100	0.991	8.70 × 10^−3^
	FC_−95%	[706.77, 102.89]	706.77	−1	10.68	100	0.991	8.58 × 10^−3^
	FC_95%	[736.55, 112.51]	736.55	−1	10.77	100	0.991	8.83 × 10^−3^

## Data Availability

Data not available due to legal restrictions.

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
