# Peer review of "A Decision-Making Algorithm for Concrete-Filled Steel Tubular Arch Bridge Maintenance Based on Structural Health Monitoring"

_materials, 2022, doi:10.3390/ma15196920_

Round 1

Reviewer 1 Report

This paper presents a decision-making algorithm for the CFST arch bridges based on the proposed evaluation model. The main contribution is to select four parameters (i.e., arch strains, pier strains, vertical displacements of arch truss, and crack widths of deck beam) as input and to compute the decision-making indices (i.e., most unfavorable maintenance time and most unfavorable parameter index of bridge maintenance system) by using the parameters. The proposed method is validated by a CFST arch bridge in China.

The paper requires extensive revise for the publication. Please refer to the following comments.

(Major comments)

1. What is the major contribution of the paper? Is the proposed method computationally efficient or reliable compared to the existing methods?

2. The field demonstration is not enough to validate the proposed method. Please consider numerical or lab-scale validation.

(Other comments)

1. The paper is too informative to understand the core contribution of the manuscript. Please make it concise for the readership. For example, the regression models in Section 4.2. need to be shorten by citing a paper because they are the well known methods.

2. Adding a nomenclature is recommended.

3. (Section 0) The literature review needs to focus on the data-driven decision-making models and their limitations to claim for contribution. Several decision-making models can be found as the following papers.

Wu, C., Wu, P., Wang, J., Jiang, R., Chen, M., & Wang, X. (2021). Critical review of data-driven decision-making in bridge operation and maintenance. Structure and Infrastructure Engineering18(1), 47-70.

Penadés-Plà, V., García-Segura, T., Martí, J. V., & Yepes, V. (2016). A review of multi-criteria decision-making methods applied to the sustainable bridge design. Sustainability8(12), 1295.

4. (Page 3 line 100) The major challenges in the existing studies are referred to as the long computation time and the inconsistency. However, the reviewer cannot find the relevant contribution in the manuscript. Is the computation time short for the proposed framework or the prediction by the proposed algorithm consistent compared to the other prediction algorithms?

5. (Figure 1) Figure 1 is not explained in the manuscript. Please consider removing it or moving to the summary section.

6. (Section 4) Provide the mechanical reason of choosing four parameters for the proposed method in Section 4.1.1.

7. (Equation 21) What is the mechanical background for the equation 21? Is this equation heuristically found by the authors?

Author Response

Sep. 10th 2022

Dear Reviewer 1#:

On behalf of my co-authors, we thank you very much for giving us the opportunity to revise our manuscript, we appreciate editors and reviewers very much for your positive and constructive comments and suggestions on our manuscript entitled “A Monitoring-based Decision-making Algorithm for Concrete-filled Steel Tubular Arch Bridge Maintenance” (ID: materials-1869547).

I am very grateful to your comments for the manuscript. Those comments are all valuable and very helpful for revising and improving our paper, as well as the important guiding significance to our researches. We have studied all of your comments carefully and have made all corrections which we hope meet your approval.

The main corrections are list in the revised manuscript according to the response to your comments, marked in red in the paper, and some of your questions were answered below. Also we hope that the revised manuscript is now suitable for publication. Looking forward to hearing from you.

Thank you again and best regard.

Yours sincerely,

Chengzhong GUI

Reviewer 2 Report

SUMMARY

The article submitted for review is devoted to a topical issue. It considers a decision-making algorithm based on monitoring for the maintenance of reinforced concrete tubular arch bridges. This is a very relevant topic for modern construction and transport facilities. In particular, the study aims to create a new heuristic algorithm for evaluating life cycle efficiency. The authors used a very modern and effective methodological apparatus, which is important to apply from the point of view of modern science and practice. The research and analytical part of the article looks very powerful. At a minimum, the authors used modeling of the ultimate bearing capacity of CFST elements, multi-parameter selection, setting limit thresholds based on the finite element method, data processing, etc. The authors obtained very important results, which showed that the width of the cracks deserves the most attention in the maintenance of concrete bridges, and the technology of such monitoring significantly increases the efficiency of monitoring the condition of existing concrete bridge structures. Thus, the article has a scientific novelty, practical significance and, in addition, is of particular interest to engineers. Thus, the article has many advantages, but at the same time, there are disadvantages, which are listed below.

COMMENTS

1.    The authors were too carried away in the abstract with a description of the research methodology. The authors described a fairly large number of iterations that they applied during the study, but paid too little attention to the results ‒ this is not entirely true. It is necessary to present the scientific result in more detail, to reflect its quantitative and qualitative characteristics. In addition, the abstract does not contain a clear formulation of the scientific problem. What did the authors decide in their study, what goals did they pursue, were they achieved and what are the percentage expressions of the result obtained, does this study differ from previously conducted, does it have scientific novelty, and how much does the study surpass the work of previous authors? Thus, the abstract should be reworked.

2.    Some overly generalized terms are also given in the keywords, for example, multi-parameter regression, parameter prediction. In addition, “concrete-filled steel tubular” looks too specific. Perhaps the keywords should have been reworked somehow in order for the article to be more correctly visible in search access.

3.    In the Introduction section, the authors did a work on the literature review on the research topic, but this work was not done very well. In particular, the authors considered only 14 sources. For modern transport construction, especially in the application of modeling methods for monitoring such structures, a fairly large number of studies on this topic are known. The Introduction should be substantially revised and the number of sources analyzed should be supplemented to at least 25. Then it will be possible to speak of scientific novelty and difference from previous studies. At the moment, the review looks haphazard.

4.    The authors, apparently, made a technical error due to the fact that the numbering of the sections was knocked down, in particular, Introduction is numbered zero, the Research objectives and framework section is numbered one, and for some reason the subsections of the first section are numbered 2.1 and 2.2. Apparently, the authors are somewhat confused in the numbering.

5.    Figure 1 should be reworked. This presentation of graphic material looks very strange. In particular, the graphs shown in these figures are not readable. If these are just screenshots of some software that the authors use, then this should be described in more detail. Figure 1 has no text after itself, section three immediately begins ‒ this is incorrect, methodologically it looks incorrect.

6.    I would like to see a more detailed interpretation of Figure 2.

7.    In general, section 3 has a somewhat protocol character, resembling more a protocol for calculating and modeling some structures than a scientific text. The interpretation text should be worked on. Approximately the same remark applies to section 4.

8.    In section 4, figure 6 attracts attention. It is interesting, but I would like to see a more detailed description of it.

9.    Table 9 is presented very ponderously and much of it is not readable. The same remark applies to Figure 10 and Figure 12. That is, the authors oversaturated the graphic component of the article, but did not strengthen the analytical part and discussion. The discussion should be a comparative analysis with the results obtained earlier by other authors, otherwise the article will not be of a scientific nature.

10.  As already mentioned, the number of literary sources should, firstly, be more than 50 titles for such an actual topic, and secondly, contain a large number of studies over the past 5 years.

11.  In general, the comment on the article is as follows: it is interesting, the study has been carried out quite large, but it should be finalized in accordance with the comments, sent for re-review. In addition, there are some remarks on the style of the English language.

Author Response

Sep. 10th 2022

Dear Reviewer 2#:

On behalf of my co-authors, we thank you very much for giving us the opportunity to revise our manuscript, we appreciate editors and reviewers very much for your positive and constructive comments and suggestions on our manuscript entitled “A Monitoring-based Decision-making Algorithm for Concrete-filled Steel Tubular Arch Bridge Maintenance” (ID: materials-1869547).

I am very grateful to your comments for the manuscript. Those comments are all valuable and very helpful for revising and improving our paper, as well as the important guiding significance to our researches. We have studied all of your comments carefully and have made all corrections which we hope meet your approval.

The main corrections are list in the revised manuscript according to the response to your comments, marked in red in the paper, and some of your questions were answered below. Also we hope that the revised manuscript is now suitable for publication. Looking forward to hearing from you.

Thank you again and best regard.

Yours sincerely,

Chengzhong GUI

Reviewer 3 Report

1. Your concept for a novel heuristic algorithm is excellent; however, predictive analysis and data acquisition are already challenging techniques for real-world SHM practise; therefore, please describe how accurate your approach is.

2. Considering that concrete-filled steel tubular (CFST) is a composite member that is difficult to monitor with sensors, could you provide a justification for selecting composite section?

3. When the crack is monitored, its width can be estimated; this has implications for the regression model. may I ask for clarification?

4. Does 86 days of monitoring compose a sufficient amount of data to perform predictive analysis?

5. How accurately are numerical model data being compared to real-time monitoring because there are numerous environmental issues that cannot be accurately simulated in numerical models? Please clarify

6. Future works contain only a few concluding points, so if at all possible, I would suggest creating a separate session for them.

Author Response

Sep. 10th 2022

Dear Reviewer 3#:

On behalf of my co-authors, we thank you very much for allowing us to revise our manuscript. We appreciate the editors and reviewers very much for your positive and constructive comments and suggestions on our manuscript entitled “A Monitoring-based Decision-making Algorithm for Concrete-filled Steel Tubular Arch Bridge Maintenance” (ID: materials-1869547).

We are very grateful for your comments on the manuscript. Those comments are all valuable and very helpful for revising and improving our paper, as well as the essential guiding significance to our research. We have studied your comments carefully and made all corrections which we hope meet your approval.

The main corrections are list in the revised manuscript according to the response to your comments, marked in red in the paper, and some of your questions were answered below. Also, we hope that the revised manuscript is now suitable for publication. We are looking forward to hearing from you.

Thank you again and best regard.

Yours sincerely,

Chengzhong GUI

Reviewer 4 Report

Generally good paper as it contains a wealth of analysis and prediction. Authors need to highlight the novelty of the work since there has been substantial amounts of research on health monitoring for bridges and others

It is too clear which data are the real dat and which is the predicted, authors need to clarify

A list of nomenclature is required (i.e symbols and abbreviations)

Title can be changed to include health monitoring

Table 1.. Previous study…present study…did the authors collect their own experimental data or it is predicted data.

Some key results on the difference between the predicted data and real data.

Conclusions are rather short. Try combining and include some values about percentage trend. Also they should relevant to the title of the paper and abstract (i.e. health monitoring, etc..)

Is it corrosion rate of reinforcement or weight change….. and concrete grades, proposed as . stands 249 for the strength degradation coefficient, considering the influence of corrosion rate Dw af- 250 ter freeze-thaw, proposed as , (1 0.0005 ( ))(1 ( 30) / 700) sd ft c c k N t f = − + − ,sr co k , 1 0.0015 ( )

English is generally good but requires improvement in places

 Nevertheless, [[[ there is the fact that an urgent need should be paid attention to for the  ]] efficient inclusion of structural health monitoring (SHM) data in structural assessment and prediction models [2].

For the [[ assessment research ]] of structural health monitoring, the SHM systems mainly focus on tracing the structural behavior and condition of the long-span bridges over their lifetime [4].

Details [[  can be encompassed the following  ]] aspects,

Author Response

Sep. 9th 2022

Dear Reviewer 4#:

On behalf of my co-authors, we thank you very much for allowing us to revise our manuscript. We appreciate the editors and reviewers very much for your positive and constructive comments and suggestions on our manuscript entitled “A Monitoring-based Decision-making Algorithm for Concrete-filled Steel Tubular Arch Bridge Maintenance” (ID: materials-1869547).

We are very grateful for your comments on the manuscript. Those comments are all valuable and very helpful for revising and improving our paper, as well as the essential guiding significance to our research. We have studied your comments carefully and made all corrections which we hope meet your approval.

The main corrections are list in the revised manuscript according to the response to your comments, marked in red in the paper, and some of your questions were answered below. Also, we hope that the revised manuscript is now suitable for publication. We are looking forward to hearing from you.

Thank you again and best regard.

Yours sincerely,

Chengzhong GUI

Round 2

Reviewer 2 Report

The authors responded well to comments and made significant revisions to the manuscript. I have no more comments.